

# Two-stage object detection in low-light environments using deep learning image enhancement

Ghaith Al-refai[1], Hisham Elmoaqet[1], Abdullah Al-Refai[2], Ahmad Alzu'bi[3], Tawfik Al-Hadhrami[4] and Abedalrhman Alkhateeb[5]

[1] Department of Mechatronics Engineering, German Jordanian University, Amman, Jordan
[2] Software Engineering Department, King Hussein School of Computing Science, Princess Sumaya University College for Technology, Amman, Jordan
[3] Department of Computer Science, Faculty of Computer and Information Technology, Jordan University of Science and Technology, Irbid, Jordan
[4] Department of Computer Science, School of Science and Technology, Nottingham Trent University, Nottingham, United Kingdom
[5] Department of Computer Science, Lakehead University, Thunder Bay, Canada

Corresponding author
Ghaith Al-refai,
ghaith.alrefai@gju.edu.jo

## ABSTRACT

This study presents a two-stage object detection system specifically tailored for low-light conditions. In the initial stage, supervised deep learning image enhancement techniques are utilized to improve image quality and enhance features. The second stage employs a computer vision algorithm for object detection. Three image enhancement algorithms—ZeroDCE++, Gladnet, and two-branch exposure-fusion network for low-light image enhancement (TBEFN)—were assessed in the first stage to enhance image quality. YOLOv7 was utilized in the object detection phase. The ExDark dataset, recognized for its extensive collection of low-light images, served as the basis for training and evaluation. No-reference image quality evaluators were applied to measure improvements in image quality, while object detection performance was assessed using metrics such as recall and mean average precision (mAP). The results indicated that the two-stage system incorporating TBEFN significantly improved detection performance, achieving a mAP of 0.574, compared to 0.49 for YOLOv7 without the enhancement stage. Furthermore, this study investigated the relationship between object detection performance and image quality evaluation metrics, revealing that the image quality evaluator NIQE exhibited a strong correlation with mAP for object detection. This correlation aids in identifying the features that influence computer vision performance, thereby facilitating its enhancement.

## INTRODUCTION

Detecting objects in low-light conditions is vital for a range of applications where visibility is compromised. For example, autonomous vehicles need to navigate effectively during nighttime and in challenging weather conditions such as heavy rain, snow, and fog (*Jhong et al., 2021*; *Liu et al., 2022*). Additionally, low-light object detection is crucial for security

systems, warehouses, factories, and search and rescue operations (*Banuls et al., 2020*). Nevertheless, the limited texture information and the presence of noise present considerable challenges in detecting objects in low-light environments, which complicates the ability of computer vision systems to accurately identify objects in such images.

Various sensors have been specifically designed for low-light detection. Infrared imaging, often combined with red, green, and blue (RGB) images, is frequently employed for night vision applications (*Deng et al., 2021*). Thermal imaging sensors function by detecting temperature variations through the measurement of infrared radiation emitted by objects (*Nguyen, Rosser & Chahl, 2021*). Light-based sensors, such as LIDAR, operate by emitting light and measuring the time it takes for the light to return after reflecting off an object (*Rashed et al., 2019*). However, these low-light sensors face several challenges, including high costs, significant computational requirements, and potential interference from other sources of radiation.

Image enhancement techniques designed to improve image quality and visualization present a promising alternative to conventional low-light vision sensors (*Qi et al., 2021*). These methods can significantly increase the visibility of objects in low-light conditions, making them an appealing choice due to their lower cost and reduced computational requirements compared to sensor-based solutions. Unlike specialized sensors, image enhancement algorithms can be implemented using standard RGB cameras, offering a more accessible and scalable approach for improving low-light imagery. Various methods exist for enhancing image visualization, including traditional techniques such as histogram equalization, gamma correction, image denoising, statistical models, and Retinex models (*Kim, 2022*). More recently, artificial intelligence techniques, particularly deep learning, have been employed to enhance the quality of low-light images (*Tang et al., 2023*).

This study introduces a two-stage object detection system. In the first stage, images undergo processing through deep learning image enhancement algorithms to improve their quality and enhance details and features. In the second stage, the enhanced images are fed into a computer vision algorithm for object detection. Three deep learning image enhancement algorithms—Zero-Reference Deep Curve Estimation (ZeroDCE++) (*Li, Guo & Loy, 2021*), Global Illumination Aware and Detail-Preserving Network (Gladnet) (*Wang et al., 2018*), and two-branch exposure-fusion network for low-light image enhancement (TBEFN) (*Lu & Zhang, 2021*)—were evaluated as part of the initial stage of this approach to enhance low-light images. For the object detection component, the YOLOv7 algorithm was employed in the second stage. The three enhancement techniques in the first stage are designed to produce images with improved color accuracy, enhanced contrast, reduced noise, and better local and global features. This results in richer feature representations for YOLOv7, ultimately leading to enhanced detection performance.

The study evaluates the quality of the enhanced images using various established no-reference evaluation metrics and examines. It also asses object detection performance of the system. The correlation between these image quality evaluators and object detection outcomes was explored. The ExDark dataset was employed to train and evaluate the detection system (*Loh & Chan, 2019*).

The image enhancement algorithms selected for this study were chosen for their unique methodologies in improving image colors, restoring lost details, and enhancing features such as texture and edges. ZeroDCE++ employs a convolutional neural network (CNN) to identify and apply light enhancement curves. Gladnet utilizes an encoder-decoder network for illumination estimation, followed by another network dedicated to detail reconstruction. TBEFN implements a dual-branch mechanism to recover distorted images and effectively combines features from both branches through a fusion approach. Furthermore, these algorithms were selected based on their outstanding performance in an evaluation of image enhancement techniques across various low-light datasets. Additional details regarding these algorithms and their performance evaluations can be found in the section on deep learning image enhancement techniques.

The article is structured as follows: 'Literature Review' provides a literature review and discusses related work in the field. 'Methodology' outlines the methodology used to develop and evaluate the two-stage object detection system in terms of both image quality and object detection performance. This section includes a concise overview of the enhancement algorithms, the computer vision algorithm (YOLOv7), the dataset, and the evaluation metrics employed. 'Results' presents the results of the study. 'Discussion' offers a discussion of the outcomes. Finally, the 'Conclusion' summarizes the conclusions and key findings of the project.

## LITERATURE REVIEW

Several methods have been developed to enhance image quality in low-light conditions. Statistical approaches, such as modifying the distribution of pixel values through histogram equalization, have been widely used (*Gonzalez, 2009*). Modified versions of histogram equalization were introduced specifically for low-light image enhancement to reduce noise (*Gu et al., 2014*). Retinex-based approaches have also been utilized for this purpose. The Retinex theory, developed by Land, was inspired by the human retina and cortex system (*Land, 1977*). A typical Retinex model-based method decomposes a low-light image into reflection and illumination components, with the estimated reflection component being treated as the enhanced result. For example, the low-light image enhancement (LIME) algorithm, a Retinex-based technique, estimates pixel illumination by identifying the maximum value in the R, G, and B channels (*Guo, Li & Ling, 2016*). *Park et al. (2017)* introduced an optimization-based low-light image enhancement method using a spatially adaptive Retinex model. Additionally, *Gu et al. (2019)* proposed a Retinex-based fractional-order variations model for enhancing severely low-light images.

In recent years, deep learning techniques have been increasingly applied to enhance low-light images, leading to notable improvements in image quality (*Tian et al., 2023*). These techniques are categorized into supervised, unsupervised, and semi-supervised approaches. Supervised learning relies on labeled training data to develop models that can then predict labels for new images. Notable examples of supervised deep learning methods for image enhancement include Low-Light Net (LLNet) (*Lore, Akintayo & Sarkar, 2017*), Global Awareness and Detail Retention Network (GLADNet) (*Wang et al., 2018*), and

Detection Transformer (DETR) (*Carion et al., 2020*). In contrast, unsupervised learning operates on unlabeled data, with EnlightenGAN standing out as a prominent unsupervised image enhancement technique (*Jiang et al., 2021*). Semi-supervised learning, which combines both labeled and unlabeled data, is exemplified by the deep recursive band network (DRBN) for image enhancement (*Qiao et al., 2021*). The image enhancement techniques used in this work all fall under the category of supervised deep learning.

Many recent studies have used deep learning enhancement techniques for low-light detection. *Wu et al. (2022)* introduced an edge-based detection stage and a cloud-based enhancement, in which a Faster R-CNN variant was used for object detection and a deep learning network was used to enhance images. In order to improve object detection in low-light pictures, *Lim, Ang & Loh (2022)* suggested a deep enhancement-object features fusion model, in which Yolov5 neck layers were fused at different stages with the Deep Lightening Network (DLN). Renitex Net with single shot detector (SSD) was the hybrid model that *Balakrishnan et al. (2024)* suggested for low light object detection. Continual learning approach was proposed for image enhancement in adverse weather conditions (*Cheng et al., 2024*).

A computer vision algorithm for object detection is essential for assessing the effectiveness of image enhancement techniques in improving detection accuracy. "You Only Look Once" (YOLO) is one of the most successful algorithms in object classification and localization due to its high detection accuracy and optimized runtime (*Jiang et al., 2022*). Over the past few years, several versions of YOLO have been released, each offering continuous improvements in both accuracy and runtime. The first three iterations of YOLO (YOLOv1, YOLOv2, and YOLOv3) were developed by *Redmon et al. (2016)*, *Redmon & Farhadi (2017)*, *Redmon & Farhadi (2018)*, followed by YOLOv4 and YOLOv7 (*Bochkovskiy, Wang & Liao, 2020*; *Wang, Bochkovskiy & Liao, 2023*). Although there are more recent versions, such as YOLOv8 and YOLOv9, YOLOv7 remains the latest official YOLO algorithm with a published scientific article. The innovations introduced in YOLOv7 to enhance detection performance make it our preferred choice for evaluating object detection on the enhanced images in the second stage of the proposed system. Several works have adopted YOLOv7 to enhance the detection in challenging environments such as *Qiu et al. (2023)* and *Hui, Wang & Li (2024)*.

Image quality evaluators (IQE) are utilized to quantitatively measure the enhancement in the image quality. They can be categorized as either full-reference (*Pedersen & Hardeberg, 2009*), which requires a reference image, or no-reference (*Mittal, Moorthy & Bovik, 2012*), which does not require a reference image for quality assessment. No-reference metrics were chosen for this work as they align with real-world scenarios where a reference image is often unavailable.

At present, the impact of deep learning-based image enhancement algorithms on the performance of advanced computer vision systems like YOLO is not well understood. Moreover, the relationship between image quality evaluation metrics and object detection performance has not been thoroughly explored. The two-stage detection system seeks to fill these gaps by analyzing how deep learning image enhancement techniques affect object

detectors performance and exploring the correlation between no-reference image quality metrics and object detection results.

## METHODOLOGY

Three versions of the two-stage detection system were designed and tested. YOLOv7 without the enhancement stage was implemented to serve as a reference model for comparison. The ExDark dataset was used for training and evaluation, no pre-processing implemented to the input dataset and it was fed to the two-stage system without modification. Figure 1 illustrates the framework of the detection system. The following subsections explore the deep learning image enhancement algorithms used in the first stage, YOLOv7 for object detection in the second stage, the ExDark dataset, and the evaluation metrics applied to both image quality enhancement and object detection.

### Deep learning image enhancement techniques

The coming subsections explore the three deep learning image enhancement algorithms used in this evaluation study.

#### Zero-reference deep curve estimation (ZeroDCE++)

Zero-DCE takes a low-light image as input and generates high-order light enhancement curves as output (*Li, Guo & Loy, 2021*). These curves are then iteratively applied for pixel-wise adjustments on the dynamic range of the input image to produce an enhanced image. The curve estimation process ensures that the range of the enhanced image is maintained while preserving the contrast of neighboring pixels. The light-enhancement curve is applied individually to the three RGB channels rather than just the illumination channel. This three-channel adjustment helps better preserve the inherent color and reduces the risk of over-saturation.

The deep network employed in this approach is referred to as DCE-Net. This network is a CNN comprising seven convolutional layers. Each layer contains 32 convolutional kernels, each with a size of $3 \times 3$ and a stride of 1, followed by the ReLU activation function. The final convolutional layer utilizes the Tanh activation function, resulting in the generation of 24 parameters. The network is designed to learn the mapping between an input image and its corresponding optimal curve parameter maps. The loss function implemented in this method is the aggregate of several components: spatial consistency loss, exposure control loss, illumination smoothness loss, and color constancy loss. The Python implementation of the algorithm is available on gitHub (https://github.com/Li-Chongyi/Zero-DCE_extension). A block diagram illustrating the operation of the ZeroDCE++ algorithm is presented in Fig. 2.

ZeroDCE++ was assessed against several enhancement algorithms, including EnlightenGAN (*Jiang et al., 2021*), RenetixNet (*Wei et al., 2018*), the Illumination Estimation-based method (LIME) (*Guo, Li & Ling, 2016*), and the Simultaneous Reflection and Illumination Estimation (SRIE) (*Fu et al., 2016*). This evaluation utilized various datasets, such as null pointer exceptions (NPE) (*Durieux et al., 2017*) and DICM

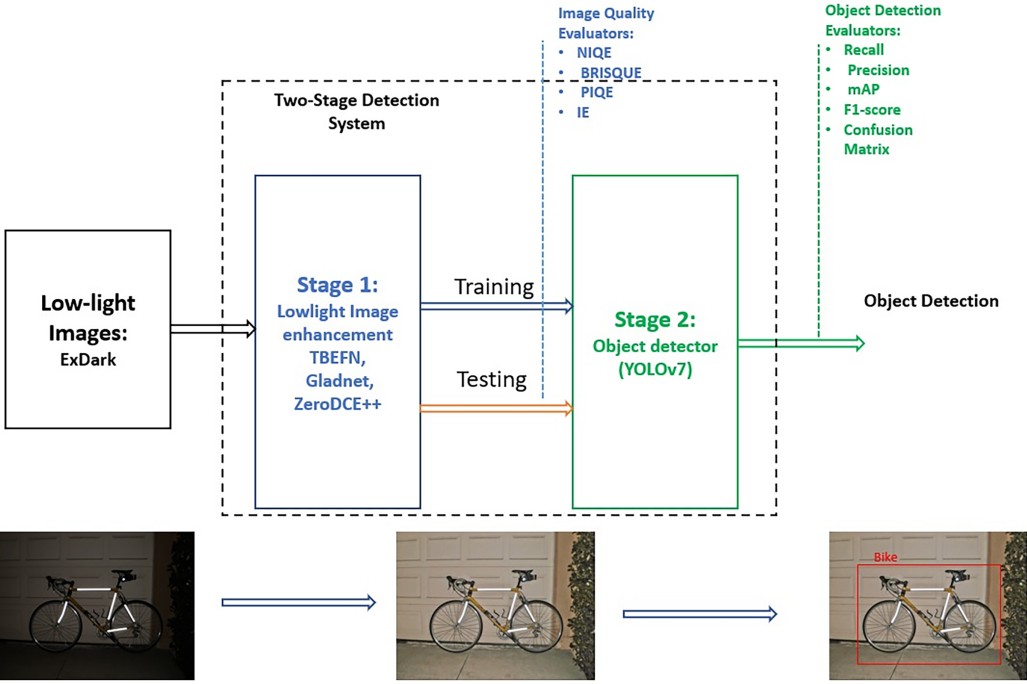

**Figure 1** The framework of the two-stage detection system and the steps utilized for evaluating image quality enhancement and object detection improvement.

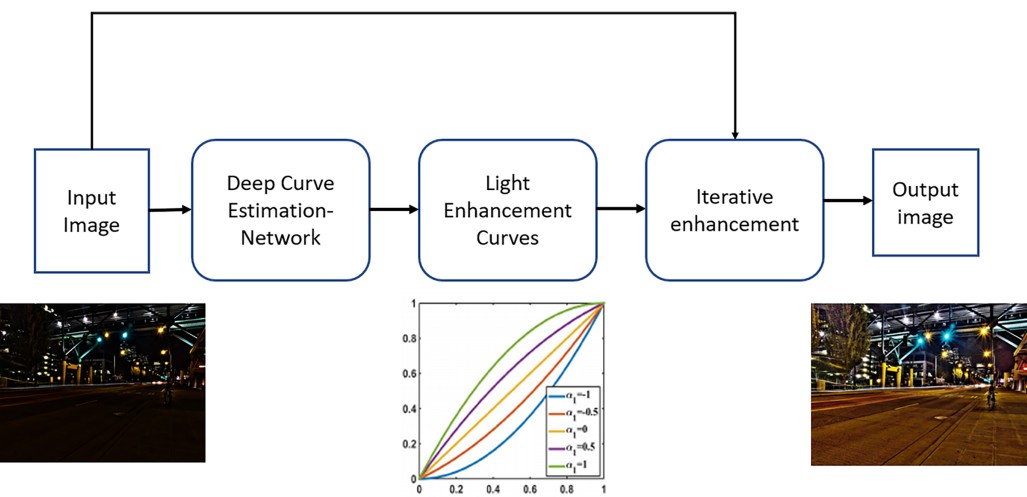

**Figure 2** The block diagram for the steps and mechanisms employed by ZeroDCE++ to achieve enhanced image quality.

(*Lee, Lee & Kim, 2013*). The findings indicated that ZeroDCE++ surpassed the other algorithms in terms of the Perceptual Index (PI) (*Mirmehdi & Perissamy, 2002*). Furthermore, the algorithm demonstrated high average precision in face detection in low-light environments (*Li, Guo & Loy, 2021*).

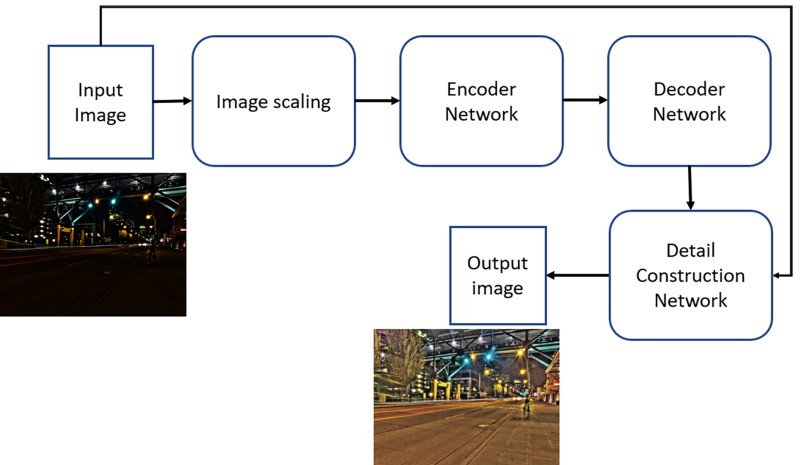

**Figure 3** **The block diagram of the Gladnet algorithm for enhanced images.**

### Global illumination aware and detail-preserving network (Gladnet)

The Gladnet algorithm operates in two distinct phases (*Wang et al., 2018*). The initial phase involves global illumination estimation, while the subsequent phase focuses on detail reconstruction. During the illumination estimation phase, the input image is resized to a specific dimension and processed through an encoder-decoder network to produce global prior knowledge regarding the illumination. An additional up-sampling block is then employed to resize the feature maps back to the dimensions of the original input image, thereby endowing the network with a global awareness of the illumination context.

In the detail reconstruction phase, the original input image is combined with the output from the global illumination estimation network, ensuring that both the original information and the illumination estimation are preserved and conveyed to the subsequent step. This integration is achieved through a concatenation layer, followed by three convolutional layers with ReLU activation functions. The loss function employed in this approach is the absolute mean difference between the output image produced by GLADNet and the corresponding ground truth image. GLADNet was trained using a synthetic dataset generated from raw images. The python implementation of the algorithm is available on GitHub (https://github.com/weichen582/GLADNet). Figure 3 provides a summary of the block diagram representing the architecture of GLADNet for low-light image enhancement.

Gladnet demonstrated superior performance compared to several enhancement algorithms based on the NIQE metric, including DeHZ (*Dong, Pang & Wen, 2010*), LIME, and SRIE (*Wang et al., 2018*). Additionally, Gladnet exhibited enhanced object detection capabilities on the improved images when evaluated using the Google Cloud Vision API.

### A two-branch exposure-fusion network for low-light image enhancement (TBEFN)

TBEFN is a deep learning-based method designed to enhance low-light images (*Lu & Zhang, 2021*). It employs a dual-branch architecture that processes images exhibiting

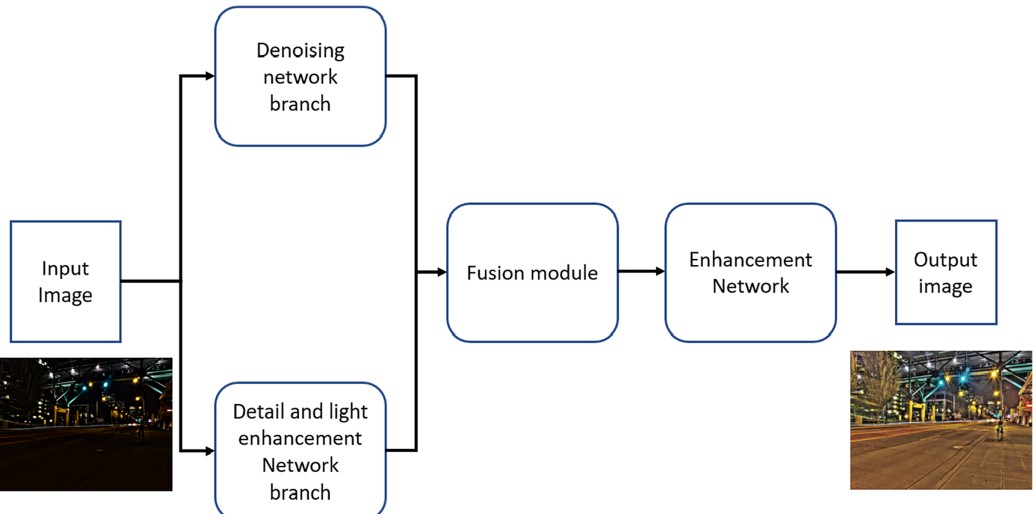

**Figure 4 The TBEFN network block diagram for low light image enhancement.**

different levels of distortion separately—one branch focuses on detail and light enhancement, while the other is dedicated to de-noising. The de-noising branch utilizes an encoder-decoder structure with skip connections to estimate transfer functions corresponding to various illumination levels. The detail and light enhancement branch consists of five convolutional layers, each with a kernel size of $3 \times 3$, with the final layer incorporating a ReLU activation function.

A generative fusion strategy is employed to effectively integrate the enhanced outputs from both branches. Furthermore, the network incorporates a self-adaptive attention unit that determines the optimal fusion weights for different regions of the image. To further improve image quality, TBEFN utilizes a final enhancement network aimed at refining the ultimately enhanced images. The loss function utilized during model training is the sum of the Structural Similarity Index Measure (SSIM) loss (*Wang et al., 2004*), VGG loss (*Ledig et al., 2017*), and smoothness loss (*Wei et al., 2018*). The pytrhon implelentation of the TBEFN is available on GitHub (https://github.com/lukun199/TBEFN). Figure 4 illustrates the network block diagram for TBEFN, specifically designed for low-light image enhancement.

TBEFN showed a leading NIQE score over other enhancing techniques such as LIME, Renetixnet and EnlightenGan in several datasets including DICM, NPE and MEF (*Lu & Zhang, 2021*).

Table 1 presents the NIQE scores for TBEFN, Gladnet, and ZeroDCE++ across three datasets, including the Exdark dataset, as reported in *Rasheed et al. (2022)*. An additional column displays the average NIQE scores derived from these results. The table illustrates that TBEFN, ZeroDCE++, and Gladnet outperform other algorithms on the Exdark dataset and achieve the highest average NIQE scores across all three datasets.

The comparison includes several efficient and widely used enhancement algorithms: SIRE (*Fu et al., 2015*), a probabilistic method that simultaneously estimates illumination

**Table 1 Comparison of enhancement algorithms across three datasets using the NIQE metric, with the algorithms organized in ascending order, where the best results are listed first (*Rasheed et al., 2022*).**

| Enhancement algorithm | LIME | DICM | ExDark | Average |
|---|---|---|---|---|
| TBEFN (*Lu & Zhang, 2021*) | 3.954 | 3.503 | 3.621 | 3.6927 |
| ZeroDCE++ (*Li, Guo & Loy, 2021*) | 3.769 | 3.567 | 3.917 | 3.751 |
| Gladnet (*Wang et al., 2018*) | 4.128 | 3.681 | 3.767 | 3.8587 |
| SIRE (*Fu et al., 2015*) | 4.050 | 3.978 | 4.383 | 4.137 |
| DHE (*Abdullah-Al-Wadud et al., 2007*) | 3.884 | 3.850 | 4.752 | 4.1620 |
| KinD (*Zhang, Zhang & Guo, 2019*) | 4.763 | 4.150 | 4.340 | 4.4177 |
| LIME (*Guo, Li & Ling, 2016*) | 4.109 | 3.860 | 4.588 | 4.1857 |
| RetinexNet (*Wei et al., 2018*) | 4.597 | 4.415 | 4.551 | 4.521 |

and reflectance in the linear domain; dynamic histogram equalization (DHE) (*Abdullah-Al-Wadud et al., 2007*), which enhances contrast based on traditional histogram equalization; Kindling Darkness (KinD) (*Zhang, Zhang & Guo, 2019*), a technique that employs deep neural networks inspired by Retinex theory to improve visibility in dark images; low-light image enhancement (LIME) (*Guo, Li & Ling, 2016*), which enhances low-light images by estimating an illumination map for each pixel to optimally adjust brightness and contrast while preserving image details; and RetinexNet (*Wei et al., 2018*), which utilizes an end-to-end trainable deep network to optimize lightness adjustment while maintaining essential image details and reducing noise.

## Object detector

The image detector employed in this study for low-light object detection is YOLOv7 (*Wang, Bochkovskiy & Liao, 2023*). YOLOv7 was selected due to its superior accuracy and speed compared to its predecessors. In a comparative study (*Zhao et al., 2024*), YOLOv7 demonstrated leading detection performance over YOLOv5, YOLOv6, YOLOv8, YOLOv9, and YOLOv10 in dark environments.

Several factors contribute to YOLOv7's enhanced performance in detecting objects in low-light conditions. The extended efficient layer aggregation network (E-ELAN) incorporates a more advanced feature extraction mechanism that effectively captures critical details. Additionally, the focal loss function utilized in YOLOv7 emphasizes the detection of hard-to-identify objects, which could be advantageous in low-light scenarios. Furthermore, YOLOv7 employs data augmentation techniques that improve the model's ability to generalize across various conditions, including challenging lighting situations. The implementation of YOLOv7 in Python is available on GitHub (https://github.com/WongKinYiu/yolov7.git).

## Dataset

Several datasets have been created for research in the field of low-light detection. For example, NightOwls dataset (*Neumann et al., 2019*) and the Dark Face dataset (*Yang et al., 2020*) focus on detecting pedestrians and faces in dark conditions. The Exclusively Dark

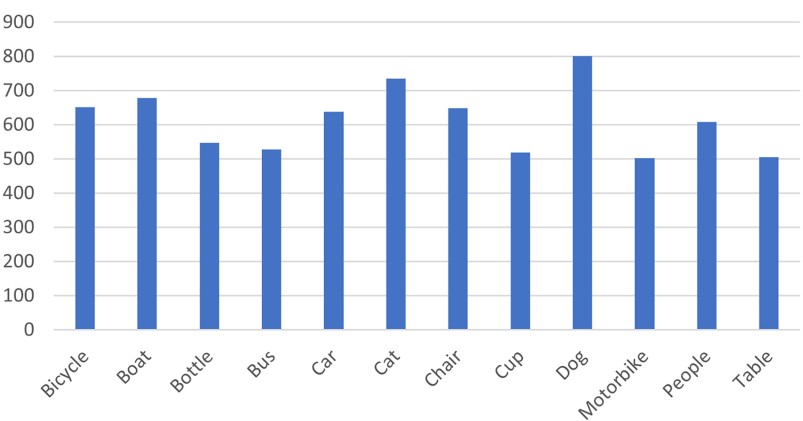

**Figure 5** The distribution of the labeled object classes in the ExDark dataset.

Image Dataset (ExDark dataset) stands out as the most comprehensive collection of low-light photographs (*Loh & Chan, 2019*). The Exdark dataset was collected and prepared by the faculty of Computer Science and Information technology, Universiti Malaya. The most recent update of the dataset has further increased its relevance at the time of authoring this study. The data is available as open source on GitHub (https://github.com/cs-chan/Exclusively-Dark-Image-Dataset.git). The ExDark dataset includes 7,363 images captured in low-light environments, annotated to encompass 12 object classes. The classes are bicycles, boat, bottle, bus, car, cat, chair, dog, motorbike, people, and table. The distribution of classes within the dataset is outlined in Fig. 5.

Three versions of the ExDark dataset were created, with each set enhanced by one of the image enhancement algorithms being evaluated. This resulted in the original ExDark dataset, a dataset enhanced using ZeroDCE++, a dataset enhanced using Gladnet, and a dataset enhanced using TBEFN. To facilitate the training and validation of the YOLOv7 model, the twelve classes in each dataset were divided into 80% for training and 20% for validation. This resulted in 5,704 images for training and 1,659 images for validation. To meet YOLOv7's annotation format requirements, a specialized code was developed to convert the original dataset annotations. This included incorporating class information, $x$ and $y$ coordinates of the bounding box center, as well as the normalized width and height of the bounding box. A special code was developed to convert the original dataset annotations format to fit YOLOv7 annotation format.

## Evaluation metrics

The evaluation metrics of this study are divided into two categories. The first category assesses the image enhancement backbones in terms of improving image quality. The second category evaluates the detection accuracy of the detector on the enhanced images. The following subsections provide an overview of the metrics used in both categories.

### Image enhancement evaluation metrics

In this work, no-reference metrics are employed to assess image quality, as the ExDark dataset does not include reference images. These metrics do not require a reference image for evaluation, making them highly applicable in real-world scenarios where a reference image may not always be available. NIQE, BRISQUE, PIQE, and IE are widely used metrics for evaluating enhanced images, as demonstrated in studies such as *Rasheed et al. (2022)*, *Tang et al. (2023)*, *Kim (2022)*, and *Guo et al. (2023)*. Additionally, the mathematical models of these metrics differ, allowing for the analysis of various features in images and their correlation with computer vision results. For instance, NIQE and BRISQUE assess image quality by measuring how much the image deviates from natural image statistics (*Mittal, Soundararajan & Bovik, 2012*; *Mittal, Moorthy & Bovik, 2012*). They primarily focus on features such as local contrast, texture patterns, and structure. PIQE focuses on local perceptual distortions, such as blur, noise, contrast, and sharpness (*Venkatanath et al., 2015*). IE measures the amount of information in an image based on pixel value distributions (*Venkatanath et al., 2015*), analyzing features such as image texture, pixel intensity variation, and image complexity. The following section provides a concise overview of the no-reference quality metrics utilized in this research.

- NIQE:

The Natural Image Quality Evaluator (NIQE) model is developed by assembling a set of quality features and fitting them to a multivariate Gaussian (MVG) model (*Mittal, Soundararajan & Bovik, 2012*). The quality features are derived from a natural scene statistic (NSS) model. The NIQE measures the distance between an MVG fit of the NSS features extracted from the test image and an MVG model of the quality features extracted from a collection of natural images. NSS extracts features such as the distribution of pixel intensities, edge orientations, and spatial frequencies. The final NIQE score is normalized to a scale ranging from 0 to 100, with lower scores indicating higher image quality.

- BRISQUE:

The Blind Referenceless Image Spatial Quality Evaluator (BRISQUE) relies on the extraction of natural scene statistics (NSS) (*Mittal, Moorthy & Bovik, 2012*). Mean-subtracted contrast normalization (MSCN) is computed for the NSS to capture neighborhood relationships. Features are then aggregated into a vector, and a trained Support Vector Machine (SVM) is utilized to assign a score to the image. Lower scores from the SVM indicate higher image quality.

- PIQE:

The Perceptual Image Quality Evaluator (PIQE) extracts various features from images, including measures related to color, contrast, sharpness, and texture (*Venkatanath et al., 2015*). The algorithm incorporates models to account for different types of artifacts, such as blur, noise, and compression artifacts, which are designed to capture the impact of these artifacts on perceived image quality. The combination of extracted features and artifact models is used to compute a quality score for the image, with higher scores indicating better image quality.

- IE:

Information Entropy (IE) was introduced by *Tsai, Lee & Matsuyama (2008)*. Information entropy indicates the amount of information the image has. Serving as a quantitative measure, the calculation of Information Entropy involves computing the normalized histogram of the input image. The IE is determined by the following equation:

$$\mathbf{IE} = \sum (NormHistogram.*log2(NormHistogram)) \tag{1}$$

### Detection evaluation metrics

The performance of the detection accuracy of the enhanced image using YOLOv7 is measured through confusion matrices (*Heydarian, Doyle & Samavi, 2022*), precision, recall, accuracy, mean average precision (mAP) and F1-score (*Powers, 2020*). The confusion matrix provides a detailed view of the system's performance, delineating the counts of true positives (TP), true negatives (TN), false positives (FP), and false negatives (FN) for each class. Precision measures the proportion of true positive predictions among all positive predictions made by the model. Recall measures the proportion of true positive predictions among all actual positive instances in the dataset. F1-score is the harmonic mean of precision and recall. mAP is the mean of the average precision of each class. The average precision is calculated as the weighted mean of precision at each threshold. The detection metrics utilized in our analysis are defined by the following equations:

$$Precision = \frac{1}{N}\sum_{i=1}^{N}\frac{TP}{TP+FP} \tag{2}$$

$$Recall = \frac{1}{N}\sum_{i=1}^{N}\frac{TP}{TP+FN} \tag{3}$$

$$F1 = 2 \times \frac{Precision \times Recall}{Precision + Recall} \tag{4}$$

$$mAP = \frac{1}{N}\sum_{i=1}^{N}AP_i \tag{5}$$

where $N$ is the number of classes and $AP_i$ is the average precision for class $i$.

## RESULTS

Pre-trained YOLOv7 weights from the COCO dataset (*Lin et al., 2014*) were used as the initial weights. Four YOLOv7 models were fine-tuned on the ExDark training datasets. The training parameters were consistent across all datasets: a batch size of 32, input images resized to 640 × 640 (the default size for YOLOv7), and training was halted at 60 epochs. The algorithms were trained and evaluated on Windows 10 computer with Nvidia GeForce GTX 1650 4 GB, 9th Gen Intel Core i7 processor, and 16 GB RAM. The utilized hardware required approximately 330 h per model for training, highlighting the limitations of using a lower-performance GPU. For similar studies, employing a higher-performance GPU is strongly recommended to significantly reduce training times.

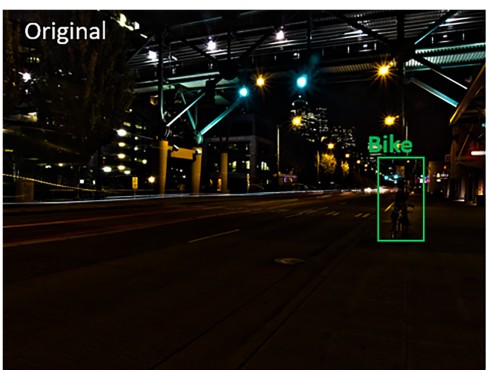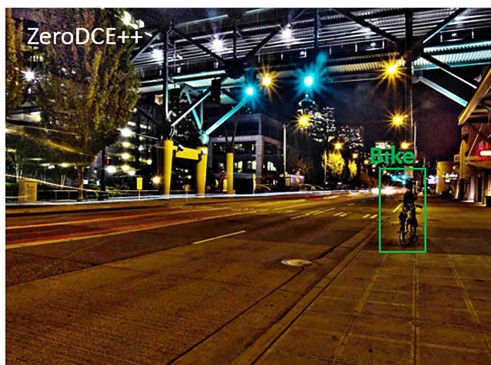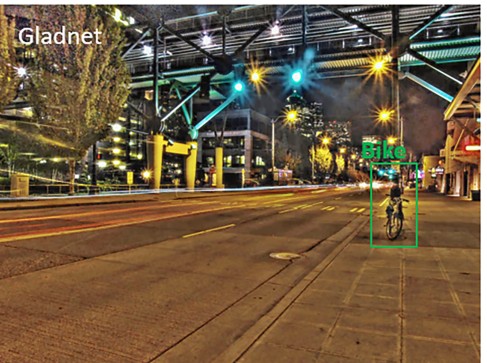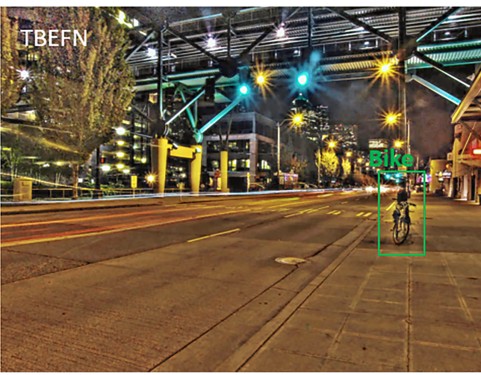

**Figure 6 An image without enhancement alongside the enhanced versions produced by ZeroDCE++, Gladnet, and TBEFN.**

**Table 2 Image quality evaluation results for the enhanced ExDark test images using ZeroDCE++, Gladnet and TBEFN with the highest scores highlighted in bold.**

| Enhancement algorithm | NIQE | BRISQUE | PIQE | IE |
|---|---|---|---|---|
| ZeroDCE++ | 3.41 | 32.58 | 40.04 | 6.83 |
| Gladnet | 3.11 | 29.57 | **38.71** | 7 |
| TBEFN | **3.05** | **28.54** | 38.84 | 6.82 |

The image quality assessment was carried out using both subjective and qualitative methods. Figure 6 provides a visual comparison between a low-light image that includes a bike and its enhanced versions generated by ZeroDCE++, Gladnet, and TBEFN. The subjective analysis suggests that TBEFN produced the most natural and detailed enhancements with with fair contrast and illumination, while ZeroDCE++ enhanced image still looks dark and the Grladnet improved color doesn't look accurate. To validate these observations quantitatively, we applied the NIQE, BRISQUE, PIQE, and IE metrics. The results, summarized in Table 2 with the best-performing results highlighted in bold. The results indicate that TBEFN delivered the best performance with NIQE and BRISQUE scores of 3.05 and 28.54, respectively, while Gladnet was close behind with scores of 3.11 and 29.57. ZeroDCE++ performed the worst, showing the highest NIQE score of 3.41 and a BRISQUE score of 32.58. However, Gladnet excelled in PIQE, achieving the best score of

**Table 3 YOLOv7 and the two-stage object detection results with ZeroDCE++, Gladnet and TBEFN as enhancing methods with the highest scores highlighted in bold.**

| Object detection method | Precision | Recall | mAP @ 0.5 | F1 score |
|---|---|---|---|---|
| YOLOv7 | 0.66 | 0.57 | 0.49 | 0.611 |
| Two-stage model (ZeroDCE++ with YOLOv7) | 0.7 | 0.62 | 0.555 | 0.657 |
| Two-stage model (Gladnet with YOLOV7) | 0.69 | **0.634** | 0.564 | 0.661 |
| Two-stage model (TBEFN with YOLOv7) | **0.73** | 0.63 | **0.574** | **0.676** |

38.71, followed closely by TBEFN at 38.84 and ZeroDCE++ at 40.04. In terms of IE, Gladnet again led with a score of 7, with ZeroDCE++ and TBEFN close behind at 6.83 and 6.82, respectively.

The enhanced images were subsequently utilized for YOLOv7 algorithm training and evaluation. Detection performance was assessed by calculating precision, recall, mAP, F1 score. This evaluation included the YOLOv7 without the stage of image enhancement. The detection results are summarized in Table 3. TBEFN combined with YOLOv7 demonstrated significant improvements in precision and mAP compared to Gladnet and ZeroDCE++. The mAP for TBEFN is 0.574, compared to 0.564, 0.555, and 0.49 for ZeroDCE++, Gladnet, and the original data, respectively.

Gladnet and TBEFN exhibited very similar recall scores, with 0.634 for Gladnet and 0.63 for TBEFN. The F1 score, which combines precision and recall, was highest for TBEFN, followed by Gladnet and ZeroDCE++. All enhancement techniques implemented in the two-stage model significantly improved detection results compared to YOLOv7. ZeroDCE ++ had the weakest detection performance among the three techniques across all evaluation metrics.

The mAP, calculated across multiple Intersection over Union (IoU) thresholds ranging from 0.5 to 0.95, was utilized to demonstrate the superiority of TBEFN. This comprehensive metric evaluates object detection performance at varying levels of localization accuracy, with the results showing TBEFN outperforming other approaches: TBEFN (0.38), Gladnet (0.366), ZeroDCE++ (0.36), and YOLOv7 only (0.31).

## DISCUSSION

The two-stage system using the TBEFN algorithm outperformed the Gladnet and ZeroDCE++ in both image quality enhancement and object detection. All three algorithms showed significant improvements over YOLOv7 without the enhacement stage.

The results revealed a strong correlation between image quality metrics NIQE and BRISQUE with object detection performance, particularly in terms of precision, mean average precision, and F1-score, as demonstrated by TBEFN. The PIQE metric was more closely associated with recall values, with Gladnet achieving the highest PIQE and recall scores. The NIQE and BRISQUE metrics reveal that the enhanced images exhibit statistical properties comparable to natural scenes, reflecting improved edge definition, balanced

**Table 4 The mAP scores for all ExDark classes using YOLOv7 and the proposed two-stage models, with the highest scores highlighted in bold.**

| Class | YOLOv7 | YOLOv7+TBEFN | YOLOv7+Gladnet | YOLOv7+ZeroDCE++ |
|---|---|---|---|---|
| Bicycle | 0.72 | 0.721 | **0.747** | 0.721 |
| Boat | 0.621 | 0.668 | **0.669** | 0.701 |
| Bottle | 0.268 | **0.304** | 0.29 | 0.289 |
| Bus | 0.597 | 0.752 | **0.787** | 0.698 |
| Car | 0.591 | **0.693** | 0.651 | 0.645 |
| Cat | 0.342 | **0.608** | 0.553 | 0.589 |
| Chair | 0.408 | 0.481 | 0.458 | **0.497** |
| Cup | 0.476 | 0.513 | 0.508 | **0.531** |
| Dog | 0.445 | **0.559** | 0.5 | 0.534 |
| Motorbike | 0.568 | **0.597** | 0.579 | 0.589 |
| People | 0.616 | **0.646** | 0.638 | 0.644 |
| Table | 0.213 | **0.335** | 0.263 | 0.323 |
| All | 0.49 | **0.574** | 0.55 | 0.564 |

contrast, and consistent texture. The E-ELAN backbone of YOLOv7 is more effective at extracting features for object detection from images with enhanced NIQE scores, benefiting from improved attributes such as edges, contrast, and texture. Furthermore, enhanced image contrast makes features more distinguishable, allowing YOLOv7's attention mechanism to more effectively identify and prioritize important features.

The IE metric demonstrated a weaker correlation with detection results. Although TBEFN had the lowest IE score, it achieved the highest detection performance. While enhanced information entropy typically indicates that images contain more details and variations, higher entropy can sometimes be associated with increased noise. This excessive noise can overwhelm the YOLOv7 model, making it challenging to differentiate between relevant features and irrelevant noise.

Table 4 presents the mAP scores for each class in the ExDark dataset using the tested algorithms. The detection results demonstrated improvements across all classes when enhancement techniques were applied compared to YOLOv7. Notably, significant enhancements were observed in the detection of cars, motorbikes, and people with the TBEFN approach, indicating its potential for autonomous driving applications. Conversely, the table category exhibited the lowest mAP score among all classes, primarily due to a high incidence of false negatives. Nevertheless, it showed the most substantial improvement when utilizing ZeroDCE++ and TBEFN.

Comparable studies have been conducted utilizing two-stage systems, such as the Multi-Branch Low-Light Enhancement Network (MBLLEN) integrated with a Faster R-CNN detector, as well as RetinexNet, which was employed alongside SSD and Faster R-CNN (*Balakrishnan et al., 2024*). The combination of TBEFN with YOLOv7 demonstrated superior detection performance compared to these previously proposed systems.

## CONCLUSION

In this study, a two-stage object detection system was introduced for low-light vision. The first stage involves implementing deep learning image enhancement technique followed by object detector in the second stage. a comprehensive evaluation was performed on three deep learning image enhancement techniques in the first stage, ZeroDCE++, Gladnet, and TBEFN. YOLOv7 was implemented as an object detector. The study utilized the ExDark dataset for both training and assessment. The effectiveness of the image enhancement techniques was measured using the image quality metrics NIQE, BRISQUE, PIQE, and IE. Concurrently, the detection performance of detection was evaluated using precision, recall, mean Average Precision (mAP), F1 score, and confusion matrices.

TBEFN demonstrated the highest image quality improvement according to NIQE and BRISQUE scores, and its PIQE score was nearly identical to that of the Gladnet algorithm. Conversely, Gladnet achieved the highest quality score based on the IE metric, while TBEFN recorded the lowest score. For the detection performance of the enhanced images using YOLOv7, TBEFN achieved the best results in terms of precision, mAP, and F1 score. Among the three techniques, ZeroDCE++ exhibited the lowest performance.

Future work could explore incorporating additional low-light datasets, particularly those tailored to specific scenarios such as nighttime autonomous vehicle driving such as Low-Light Multi-Object Tracking (LMOT) dataset (*Wang et al., 2024*) and Low Lighting Dash Cam Video dataset (*Wang et al., 2024*). Additionally, integrating image enhancement algorithms directly into the YOLO framework, rather than using a separate enhancement algorithm in conjunction with YOLO, could significantly improve detection speed and efficiency. For example, a custom layer could be designed within the YOLOv7 architecture that emulates the operations of TBEFN by dividing the input into two branches. Each branch would apply distinct convolutional operations or transformations, which would subsequently be merged. Furthermore, knowledge distillation (KD) techniques could be employed in the model to develop a low-light detection system that is both faster and requires fewer computational resources (*Cheng et al., 2024*). Integrating capsule networks (CapsNets) in the model can improve feature representation, robustness to variations, and generalization capabilities (*Liu et al., 2024*).

### Funding
The authors received no funding for this work.

### Competing Interests
The authors declare that they have no competing interests.

### Author Contributions
- Ghaith Al-refai conceived and designed the experiments, performed the experiments, analyzed the data, performed the computation work, prepared figures and/or tables, authored or reviewed drafts of the article, and approved the final draft.

- Hisham Elmoaqet analyzed the data, performed the computation work, prepared figures and/or tables, authored or reviewed drafts of the article, and approved the final draft.
- Abdullah Al-Refai performed the computation work, prepared figures and/or tables, and approved the final draft.
- Ahmad Alzu'bi performed the computation work, prepared figures and/or tables, authored or reviewed drafts of the article, and approved the final draft.
- Tawfik Al-Hadhrami performed the computation work, prepared figures and/or tables, and approved the final draft.
- Abedalrhman Alkhateeb performed the computation work, prepared figures and/or tables, and approved the final draft.

## Data Availability

The ExDark dataset is available at GitHub: https://github.com/cs-chan/Exclusively-Dark-Image-Dataset.git.

The YOLOv7 code is available in the Supplemental File.

## Supplemental Information

Supplemental information for this article can be found online at http://dx.doi.org/10.7717/peerj-cs.2799#supplemental-information.

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
