# Peer review of "Two-stage object detection in low-light environments using deep learning image enhancement"

_PeerJ Computer Science, doi:10.7717/peerj-cs.2799_

## Round 0.1 · original submission · Major Revisions

Please respond to the detailed comments from the reviewers.

·

Basic reporting

1. While the manuscript is well-written, some sentences lack clarity and could benefit from more precise wording. For example, in the abstract, phrases such as “provide insights into optimizing enhancement techniques for improved computer vision outcomes” could be rephrased to be more concise and specific. A thorough language review is recommended.

2. While the figures included are helpful, some captions are insufficiently descriptive. For instance, Figure 6 compares image enhancements but lacks explicit annotations or arrows highlighting the differences. Adding such details could enhance understanding.

Experimental design

1. Although the methodology is described in a structured manner, the preprocessing of datasets (or lack thereof) is not thoroughly discussed. Clarifying how the raw ExDark dataset was handled before feeding it into the model could improve reproducibility.

2. The study evaluates image quality using multiple metrics but does not justify why these specific no-reference metrics (NIQE, BRISQUE, etc.) were chosen over others. Including reasoning for these choices and how they align with the study goals would add rigor.

3. The experiments were conducted on a system with limited computational power (Nvidia GTX 1650). It would be helpful to discuss how this may have impacted the results, particularly for training and inference times, and whether this setup aligns with real-world use cases.

Validity of the findings

1. The manuscript does not provide confidence intervals or statistical tests to validate the superiority of TBEFN over other enhancement algorithms. Adding such analyses would strengthen the conclusions.

2. The correlation between image quality metrics and detection performance is mentioned but not deeply analyzed. For example, why does NIQE correlate strongly with mAP while IE does not? Exploring these results in greater depth could provide actionable insights for future research.

3. The findings are based solely on the ExDark dataset. It would be valuable to discuss the generalizability of the proposed method across other datasets or in practical low-light scenarios (e.g., autonomous driving).

Additional comments

1. The manuscript mentions that integrating the image enhancement process into the YOLO framework could improve runtime. Including preliminary work or hypotheses on this integration would strengthen the discussion.

2. While the YOLOv7 implementation link is provided, it is unclear whether the enhancement algorithm codes are accessible. Sharing the complete codebase could significantly improve reproducibility.

Reviewer 2 ·

Basic reporting

In the manuscript, a two-stage object detection system is proposed in order to improve the accuracy and efficiency of detecting objects under low-light conditions. The method integrates supervised deep learning image enhancement techniques in the first stage to improve image quality and enhance features, followed by leveraging a computer vision algorithm for object detection in the second stage. The proposed method has been experimentally verified with good results. However, according to the manuscript content, there are still some issues that need to be addressed:
1.The manuscript presents a target detection system combining image enhancement and YOLOv7, describing the selection and application of three deep learning image enhancement algorithms: ZeroDCE++, Gladnet and TBEFN. However, information on the specific implementation details of these algorithms, such as network architecture and loss function design, is more sketchy. The analysis of how to specifically improve the existing methods and the comparison with previous work could be further enhanced. It is recommended that the unique advantages of these several image enhancement techniques be discussed in more detail and that more citations of relevant literature be added.
2.The manuscript mentions that YOLOv7 was chosen as the target detector because of its superior performance in terms of accuracy and speed. It is suggested to further discuss why YOLOv7 is particularly suited to the low-light environment of this study and what makes it unique compared to other versions. In addition, the specific contribution of the new features introduced in YOLOv7 to enhance detection performance in low-light conditions could be explored.
3.The approaches presented in “Pixel Distillation: Cost-flexible Distillation across Image Sizes and Heterogeneous Networks” and “Continual All-in-One Adverse Weather Removal With Knowledge Replay on a Unified Network Structure” have been demonstrated to be able to improve the detection accuracy in poor image condition, such as low resolution and low quality. So, the authors should provide an indepth discussion to those work to illustrate the advantages of this work.
4.Before submitting your paper, please check the formatting and layout layout of the paper. Make sure the style of the manuscript meets the requirements. Such as the figure notes in Figure7 and the layout in Figure8.
5.In addition to the ExDark dataset, it is recommended that more types of low-light environmental datasets be added for testing, especially those containing complex backgrounds or extreme lighting conditions, in order to validate the model's ability to generalize.
6.The manuscript compares the difference in effectiveness between using YOLOv7 alone and after combining different image enhancement techniques. In order to make the comparison fairer and more reasonable, it is suggested to add a control group experiment using only the traditional image enhancement method as a pre-processing step, so as to better highlight the superiority of the proposed two-stage system.
7.The conclusion section briefly mentions some limitations and directions for future research, but is slightly thin on substance. It is recommended that the shortcomings of the current methodology be more fully summarized and that more specific and clear recommendations for improvement be made. For example, the hierarchical relationships in the scene may help to parsing the object detection, which could be solved by the CapsNets as in “Capsule_Networks_With_Residual_Pose_Routing” and “Part-Object Relational Visual Saliency”.

Experimental design

see Basic reporting

Validity of the findings

see Basic reporting

Additional comments

none

---

## Round 0.2 · accepted · Accept

Thank you for incorporating the changes required by the reviewers. The mAP values and the class wise improvement show the robustness of the methods used. As such, I agree with the reviewers that the paper is ready for publication.

·

Basic reporting

No comment

Experimental design

No comment

Validity of the findings

No comment

Additional comments

Nil

Reviewer 2 ·

Basic reporting

The authors have addressed my concerns well. Now, the manuscript looks ready for publication.

Experimental design

no comment

Validity of the findings

no comment